# Identical folds used for distinct mechanical functions of the bacterial flagellar rod and hook

Takashi Fujii[1,2], Takayuki Kato[1], Koichi D. Hiraoka[1], Tomoko Miyata[1], Tohru Minamino[1], Fabienne F. V. Chevance[3], Kelly T. Hughes[3] & Keiichi Namba[1,2]

The bacterial flagellum is a motile organelle driven by a rotary motor, and its axial portions function as a drive shaft (rod), a universal joint (hook) and a helical propeller (filament). The rod and hook are directly connected to each other, with their subunit proteins FlgG and FlgE having 39% sequence identity, but show distinct mechanical properties; the rod is straight and rigid as a drive shaft whereas the hook is flexible in bending as a universal joint. Here we report the structure of the rod and comparison with that of the hook. While these two structures have the same helical symmetry and repeat distance and nearly identical folds of corresponding domains, the domain orientations differ by ∼7°, resulting in tight and loose axial subunit packing in the rod and hook, respectively, conferring the rigidity on the rod and flexibility on the hook. This provides a good example of versatile use of a protein structure in biological organisms.

[1] Graduate School of Frontier Biosciences, Osaka University, 1-3 Yamadaoka, Suita, Osaka 565-0871, Japan. [2] Riken Quantitative Biology Center, 1-3 Yamadaoka, Suita, Osaka 565-0871, Japan. [3] Department of Biology, University of Utah, Salt Lake City, Utah 84112, USA. Correspondence and requests for materials should be addressed to K.N. (email: keiichi@fbs.osaka-u.ac.jp).

The bacterial flagellum is a motile organelle that enables bacteria to propel themselves towards favourable and away from unfavourable environments[1–5]. The flagellum is made of three distinct parts: the basal body, which functions as a rotary motor as well as a protein export apparatus; the filament, a long helical propeller that propels cell locomotion in viscous environments; and the hook, which connects the filament to the motor as a universal joint to transmits motor torque to the propeller oriented off-axis of the motor. The basal body is a large protein complex of about 8 MDa, is made of four ring complexes, and spans both the cytoplasmic and outer membranes[6,7]. In Gram-negative bacteria, such as *Salmonella enterica* and *Escherichia coli*, the basal body can be divided into five parts according to their functions: the C ring is the cytoplasmic ring with a diameter of 45 nm and is involved in torque generation as well as regulating the directional switching of motor rotation; the MS ring is the cytoplasmic membrane ring with the M ring in the membrane and the S ring in the periplasm and works as the rotor as well as the base for entire flagellar assembly; the LP ring is the bushing to support fast rotation of the rod through the peptidoglycan layer and the outer membrane; the flagellar type III protein export apparatus forms the central gate that selectively translocates flagellar axial proteins from the cytoplasm into the central channel of the flagellum; and the rod is the most proximal portion of the flagellar axial structure that works as a drive shaft (Fig. 1)[8].

The filament is a helical tubular assembly of a single protein, FliC or flagellin, with a diameter of 10–20 nm, and a few tens of thousands of subunits build a helical propeller that grows up to 10–15 µm long. Its ability to switch between polymorphic, left- and right-handed supercoiled forms plays an important role in dynamically switching its helical pitch and handedness in response to mechanical force by the reversal of the motor rotation to change the swimming mode of bacteria between run and tumble for chemotaxis[9].

The hook is also a helical tubular assembly of a single protein, FlgE, with a diameter of about 18 nm. Approximately 130 copies of FlgE subunits form a hook of about 55 nm long, and its length

is controlled within about 10% by the substrate specificity switch of the export apparatus. The hook grows directly on the rod while the hook-filament connection is through two junction proteins, FlgK and FlgL. The unique packing interactions of FlgE subunits in the hook realize the bending flexibility and twisting rigidity at the same time for the hook to work as a universal joint, allowing the motor to drive the off-axis rotation of the filament[10–12].

The rod is a relatively thin, hollow but rigid straight tube assembled on the MS ring. The maximum diameter is about 13 nm in the distal part surrounded by the LP ring, but it is 6–7 nm in the proximal part deeply inserted into the conical pipe structure of the FliF ring above its S ring part, with a total length of about 25 nm. The rod is responsible for the stable high-speed rotation of the motor driving filament rotation up to around 300 Hz in *S. enterica* and *E. coli*[13,14] and 1,700 Hz in Marine *Vibrio*[15]. Its rigidity assures the stable axial position of rotation even against its quick reversal that occurs within 1 ms. The rod rotates smoothly at high speed within the LP ring without any lubricants between their contact surfaces. The rod consists of four different proteins, FlgB, FlgC, FlgF and FlgG, with an adapter protein, FliE, for their assembly on the MS ring[16,17]. FlgB and FlgC are 15 and 14 kDa proteins, respectively, and form the thin proximal rod structure of about 4.5 nm in length and 6.0 nm in diameter. FlgF and FlgG are 26 and 29 kDa proteins, respectively, and together form the thicker, distal portion of the rod with about 20 nm in length and 13 nm in diameter. FliG is the one that forms the most distal part[18]. The stoichiometries of FlgB, FlgC, FlgF and FlgG have been estimated biochemically to be about 6, 6, 6 and 26, respectively[19]. *In vitro* studies on these rod proteins expressed and purified from *E. coli* overproduction constructs revealed that FlgB, FlgC and FlgG tend to aggregate to form β-amyloid-like fibrils that are structurally unrelated to the rod formed *in vivo*[20]. Such difficulties in handling the rod proteins impeded their structural analysis by X-ray crystallography.

The flagellar axial proteins share a common structural motif of linearly connected multiple domains and a relatively elongated shape. In the axial structures, such protein domains are arranged in the radial direction. For instance, the four domains D0, D1, D2 and D3 of flagellin and the three domains D0, D1 and D2 of FlgE are arranged radially from the inner to the outer part of the tubular structure of the filament and hook, respectively[12,21,22]. The axial proteins also share high similarity in their primary sequences, suggesting a common evolutionary origin. The N- and C-terminal chains of about 100 residues together are highly conserved and contain a heptad repeat of hydrophobic residues, and secondary structure predictions show that they form an α-helical coiled coil[16,20]. In fact, the structures of the filament and hook show that the N- and C-terminal chains of FliC and FlgE form an α-helical coiled-coil in their inner core domain D0 (refs 12,23).

The rod and hook have distinct mechanical properties, with the rod being straight and rigid and the hook being supercoiled and flexible in bending, for their functions as a drive shaft and a universal joint, respectively. Interestingly, however, the amino acid sequence of entire FlgG has a high degree of identity (39%) with the N- and C-terminal regions of FlgE[24] that form its domains D0 and D1, which are both responsible for the intersubunit interactions for hook assembly as well as for the hook to be flexible[10]. It is likely from the sequence identity that the folds of these two domains would also be nearly identical between FlgG and FlgE. How then FlgG subunits build the rigid tubular structure of the rod, even with its smaller diameter than the hook?

The hook-basal body (HBB) structure has been revealed by electron cryomicroscopy (cryoEM) and image analysis (Fig. 1) but the rod structure is shown only as a featureless cylinder

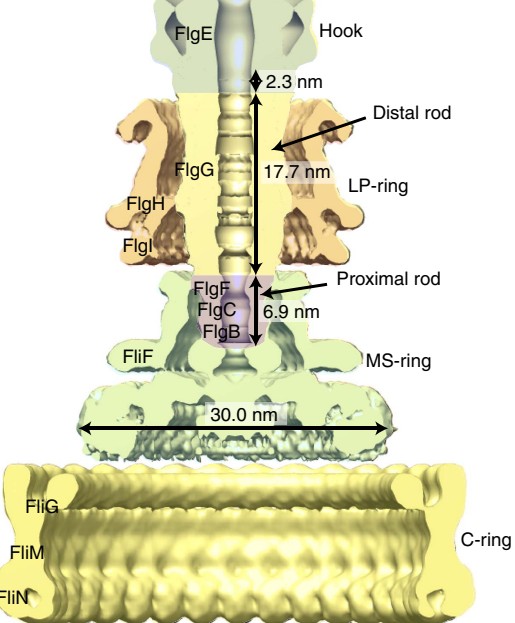

**Figure 1 | The structure of the flagellar basal body of *Salmonella enterica*.** A half-cut section of a cryoEM 3D density map is shown with its parts in different colours and with dimensions[8].

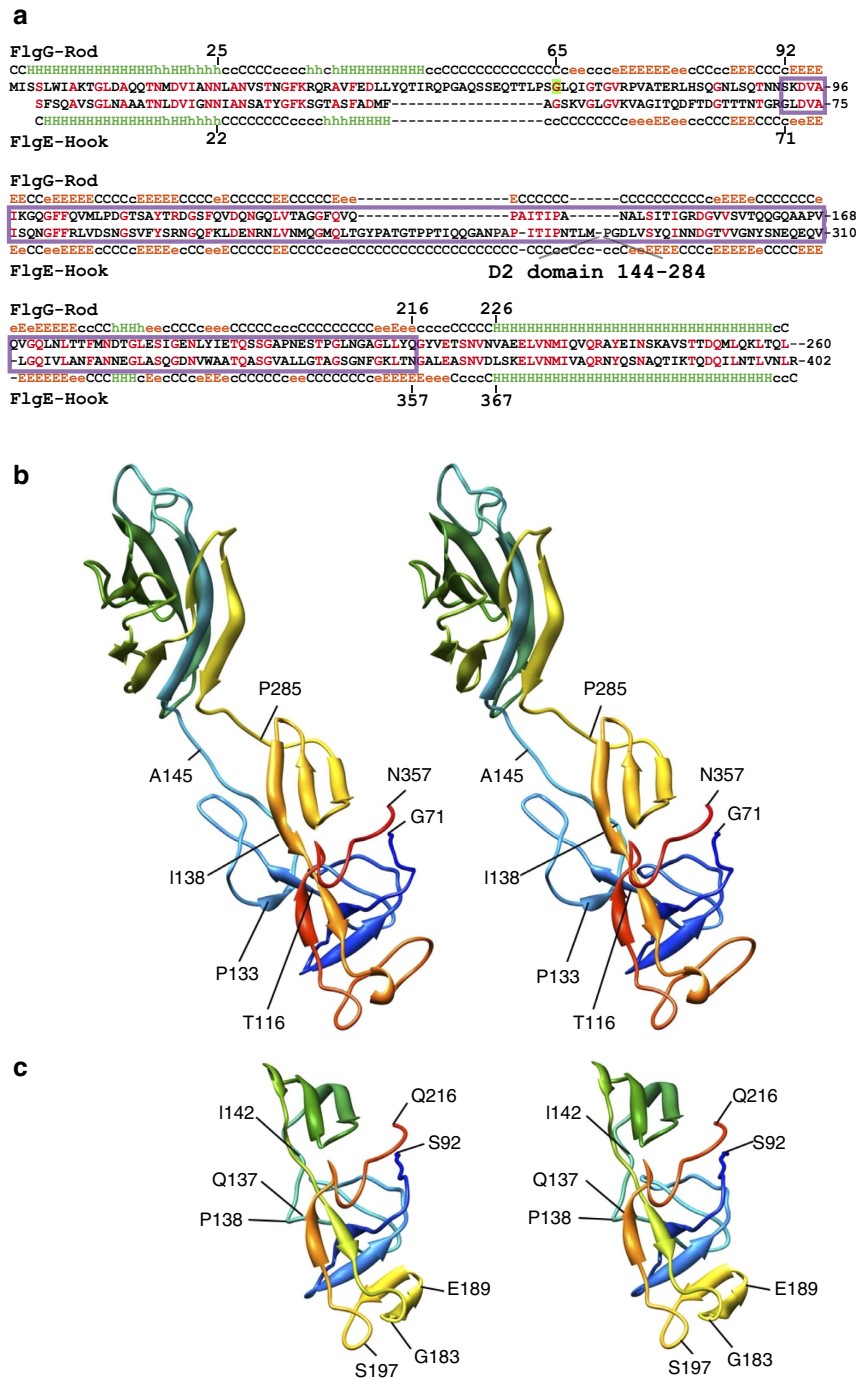

**Figure 2 | Homology modelling of FlgG based on the known crystal structure of FlgE.** (**a**) Structure-based sequence alignment of FlgG and FlgE. Purple box indicates residues 71–357 of FlgE and residues 92–216 of FlgG, which were used to build a homology model of domain D1 of FlgG. The FlgG sequence has an 18 amino-acid residues insertion after residue 43 of FlgE. A FlgG mutant with a point mutation G65V (shaded in green) is used to produce the polyrod for structural analysis. (**b**) Stereo view of the crystal structure of FlgE$_{71-357}$. (**c**) Stereo view of the model structure of the FlgG$_{92-216}$. The chains are coloured from blue to red through the rainbow spectrum from the N to C terminus.

without any sign of helical symmetry or subunit packing due to the low resolution around 20 Å and a specific rotational symmetry used for three-dimensional (3D) image reconstruction[8,25,26]. Because the rod is short and is surrounded by the LP-ring having a different symmetry from the rod, its high-resolution structural analysis within the HBB is difficult. So a long rod mutant would be desirable for structural analysis, and some point mutations in FlgG have been found to make the rod to grow as long as 1 μm (ref. 24). Since many of these

polyrods have the hook attached to their distal ends, the structure and packing interactions of the FlgG mutant subunits in the polyrod structure must be nearly identical to those in the native rod structure built with wild-type FlgG[24]. Here, we obtained a ~7 Å resolution density map of the polyrod by cryoEM helical image analysis and built an atomic model of the distal FlgG rod by docking a homology model of a core fragment of FlgG into the density map and also by modelling the terminal helices. What we have found is that only a subtle difference in the orientations and

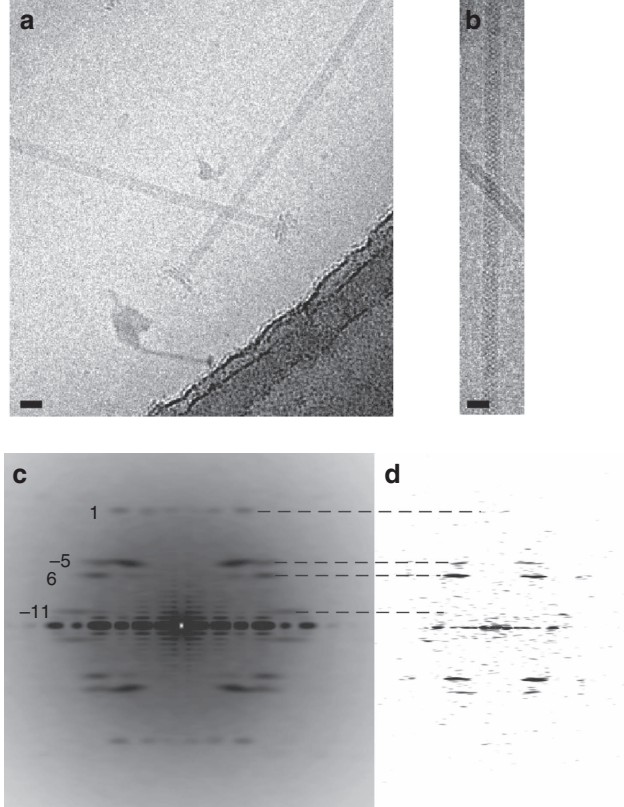

**Figure 3 | Electron cryomicrographs of the polyrod and polyhook and their Fourier transforms.** (**a**) CryoEM image of frozen-hydrated polyrods. (**b**) CryoEM of frozen-hydrated polyhooks, which were made straight by incubating them at 4 °C. Scale bars: 200 Å. (**c**) Averaged Fourier transform of the polyrod computed from ~3,000 image segments of 524 Å in length. Four layer lines are labelled with their Bessel orders. The positions of these layer lines are: 1-start, 24.0 Å$^{-1}$; 5-start, 41.0 Å$^{-1}$; 6-start, 52.2 Å$^{-1}$; 11-start, 192 Å$^{-1}$. (**d**) A Fourier transform of the polyhook computed from a single image segment. The layer line positions are almost the same as those of the polyrod.

intersubunit interactions of corresponding domains of FlgG and FlgE with nearly identical folds is responsible for the distinct mechanical functions of the rod and the hook.

## Results

**Homology modelling of FlgG.** In the axial tubular structure, protein domains are arranged radially. The hook protein FlgE has three domains, D0, D1 and D2. Domain D2 lies around 7.5 nm from the centre and forms the surface of the hook with an outer diameter of 18 nm. Domain D1 is just inside domain D2 and lies between 5 and 6 nm from the centre. Domain D0 forms a tube with a 1-nm thick wall and the axial central channel of about 2 nm in diameter. The crystal structure of a 31 kDa FlgE fragment containing domains D1 and D2 has been solved by Samatey *et al.*[10] (Fig. 2b). The amino acid sequence of FlgG has been shown to have a high degree of identity (39%) with the D0 and D1 domains of FlgE[16,24]. However, there are two significant differences between FlgG and FlgE. First, the N terminus of FlgG is longer than that of FlgE by 3 residues, and FlgG has an 18 residues insertion at residues 46–65 where the majority of polyrod mutations are located (Fig. 2a). Second, FlgE has two insertions that are not present in FlgG: 18 residues (G117–A134) forming a triangular loop in domain D1 and a central stretch of 146 residues (T141–G286) forming domain D2. We built a model

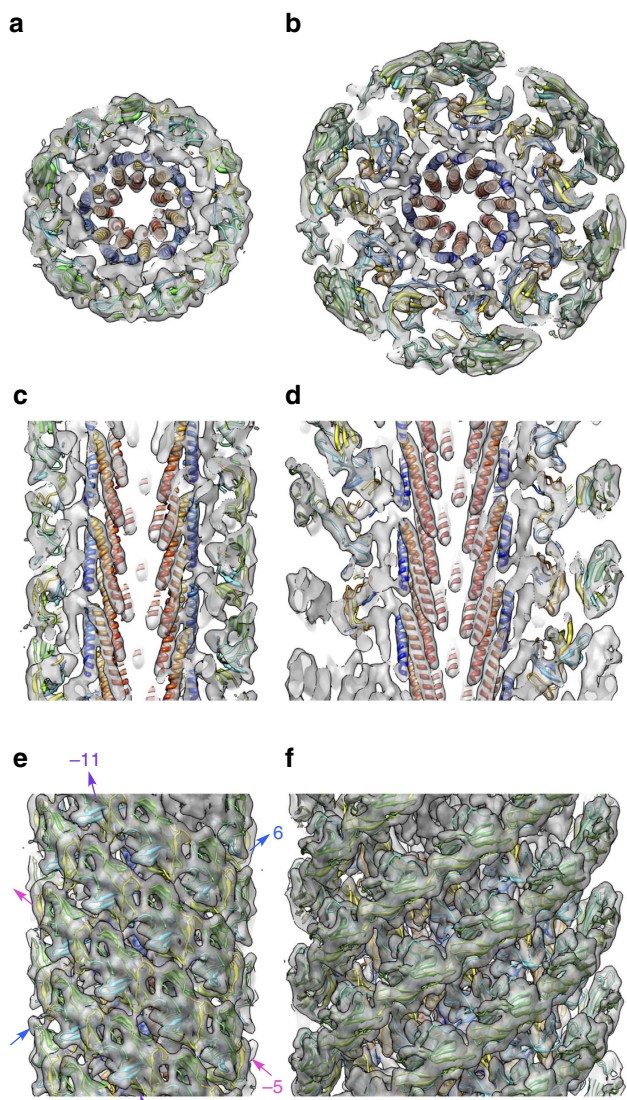

**Figure 4 | Reconstructed 3D images of the rod and the hook with fitted atomic models.** (**a,c,e**) 3D density maps of the polyrod; (**b,d,f**) 3D density maps of the polyhook. (**a,b**) end-on view of a thick cross section; (**c,d**) side view of a thick longitudinal section through the axis; (**e,f**) side view of the surface. Domain D0, D1 and D2 are labelled in **c,d**. Major helical lines are indicated by arrows and numbers: 6-start, blue; 5-start, magenta; 11-start, purple, where negative numbers indicate left-handed helices. The N- and C-terminal α-helices, which form the inner core domain D0, are coloured blue and red, respectively. The density between domains D0 and D1, which is not labelled here, was designated as domain Dc in the hook structure[12]. Scale bar, 50 Å.

of domain D1 of FlgG (residues 91–222) by homology modelling based on the structure of domain D1 of FlgE corresponding to residues 71–116, 135–140 and 287–357 by omitting the triangular loop (residues 117–134) and domain D2 (residues 141–286). We constructed a homology-based atomic model of FlgG by MODELLRv9.7 software[27]. We found the homology model of FlgG fragment to retain the main chain structure of domain D1 of FlgE (Fig. 2b,c).

**Structural analysis of the polyrod by cryoEM image analysis.** The diameter of the polyrod is 13 nm, much smaller than those of the flagellar filament (23 nm) and the hook (18 nm) (Fig. 3a,b). In addition, EM observations of negatively stained or ice-embedded

frozen-hydrated polyrods (Fig. 3a) showed images of a simple cylinder with smooth and featureless surface, unlike the filament and the hook (Fig. 3b). The Fourier transform of individual cryoEM images of the polyrod mostly showed no layer lines except for occasionally just one with an axial spacing of 40.4 Å, and therefore it was not possible to determine the helical symmetry and axial repeat distance of the subunit arrangement from individual polyrod images. However, by averaging Fourier transforms of box-segmented images, four layer lines clearly showed up (Fig. 3c), and these layer line positions were almost the same as those of the hook (Fig. 3c,d). Since the FlgG rod and the hook are directly connected without any junction proteins as an adapter, it is reasonable for them both to have the same helical symmetries and axial repeat distances. This allowed us to determine the approximate helical symmetry and axial repeat of the polyrod for further image analysis.

We determined the structure of the polyrod by cryoEM image analysis using the iterative helical real space reconstruction (IHRSR) algorithm[28]. Because the layer line positions were almost the same between the hook and rod, we used the helical symmetry and axial repeat of the hook: an axial rise of 4.12 Å and a unit azimuthal rotation of 64.78° along the right handed 1-start helix[12], as the initial value for the iterative refinement in the 3D image reconstruction by IHRSR. A solid cylinder was used as the initial reference density volume to avoid any bias toward the initial reference structure. The helical parameter was converged to an axial rise of 4.13 Å and an azimuthal rotation of 64.75°. The resolution of the final 3D image reconstruction was 7.4 Å (at Fourier shell correlation (FSC) = 0.5). The density map showed radially arranged two domains, D0 and D1, with domain D0 facing the central channel with a diameter of 13 Å and spanning a radial range of 6.5–30 Å, and domain D1 forming the smooth outer surface of the polyrod with a diameter of 125 Å (Fig. 4a,c,e).

**Atomic model of the polyrod.** We built an atomic model of the polyrod by docking the homology model of FlgG$_{91-222}$ into the cryoEM density map by using the real-space correlations in Chimera[29] and by applying the helical symmetry. The model fitted well as a rigid body into the density map, to the level of secondary structures, such as β-sheets, β-hairpins and loops, indicating that the main chain folding within domain D1 does not significantly change upon polymerization to form the rod structure (Figs 4 and 5).

The density map also resolved two rod-like densities in the inner core of the polyrod just as those in domain D0 of the hook (Figs 4a–d and 5). One of them facing the central channel is 4.8 nm long and the other is 3.5 nm long. They are likely to represent the terminal α-helices, with the longer one corresponding to ∼35 residues and the shorter one to ∼26 residues. The secondary structure prediction of the terminal sequence of FlgG predicts the C-terminal α-helix to be 33 residues long and the N-terminal one to be 23 residues long. The distinct difference in their lengths allowed us to unambiguously identify that the longer rod-like density facing the central channel is the C-terminal α-helix and the shorter one inside is the N-terminal one. The sequences of these terminal chains of FlgG have heptad repeats suggesting the presence of α-helical coiled-coils, in the same manner to the eight other flagellar axial proteins[16,20]. We placed these two terminal α-helices in the density map with their azimuthal orientations according to the relative positions of hydrophobic residues in these heptads. The residues we used are Leu-5, Leu-12, Met-19, Leu-26 for the N-terminal chain and Val-226, Met-233, Tyr-240, Val-247, Leu-254 for the C-terminal chain (Fig. 2a), and we placed the two α-helices with these

hydrophobic residues facing each other, just like those in the atomic model of FliC in the filament.

We thus built an atomic model of the FlgG rod structure with residues 1–23 (D0), 91–222 (D1) and 227–260 (D0), but we could not complete the model by filling the chains in the two gaps (24–90 and 223–226) into domain Dc (Fig. 5a), which connects domains D0 and D1, because the resolution of our present density map was not high enough to allow *de novo* model building of these two gap regions.

**Structural comparison of the rod and hook.** Although the rod (13 nm) is thinner than the hook (18 nm) by the absence of domain D2 of the hook, which forms the right-handed six-stranded continuous helical densities on the surface to stabilize the hook structure, EM observations of the polyrods and polyhooks on negatively stained specimen grids indicate that the polyrod is quite rigid against bending, much more rigid than the polyhook. The atomic model of the rod clearly indicate that the rod is rigid because the D0 domains and the D1 domains are both highly packed in all the three main helical directions: the left-handed 5-start, the right-handed 6-start and the 11-start (protofilament) helix, in each of the inner and outer radial regions of the rod, respectively (Fig. 4c,e). In contrast, although the packing interactions of the D0 and D1 domains of FlgE in the hook are also extensive in each of their radial regions, their axial packing interactions have small gaps to allow axial compression and extension of its protofilaments, thereby conferring the bending flexibility on the hook to work as a universal joint[12]. The contribution of the D2 domains of the hook to its bending rigidity is negligible because there is a large gap between each of the right-handed 6-start helical density strands on the hook surface.

It is puzzling how the FlgG and FlgE subunits can form such significantly distinct packing interactions in the rod and hook, respectively, to give rise to their markedly different mechanical properties, despite their nearly identical folds, helical symmetries and repeat distances. What we found in the structural comparison are the following two factors. The first is a slightly longer N-terminal α-helix of FlgG than that of FlgE (26 residues for FlgG and 24 residues for FlgE), and the presence of an extra density in the rod structure in the yet un-modelled region of the rod and hook density maps (domain Dc), which are formed mainly by their N-terminal regions connecting the N-terminal α-helix and domain D1. This extra density is likely to be formed by the 18 residues insertion in FlgG (Fig. 2a). The extra density and the longer N-terminal α-helix of FlgG connect axially neighboring subunits in the rod structure, but no such connection is made in the hook structure because of the shorter N-terminal α-helix and the absence of the extra density. These differences result in the tight axial packing of the N-terminal α-helices in the rod and about a 5 Å axial gap in the hook. The second one is the difference in the orientations of their D1 domains. The D1 domains of FlgG are oriented more upright to form the tight axial contacts, whereas those of FlgE are more tilted from the axis than those of FlgG by about 7°, giving rise to an axial gap of a few Å along the protofilament (Fig. 6a,b). The resolutions of 3D maps of the polyrod and polyhook are both high enough to allow accurate determination of domain orientations because the atomic models are fitted into their density maps as rigid body to restrain the stereochemistry of their domain conformations and yet all the β-sheets, β-hairpins and loops are nicely fitted into corresponding densities to validate the accuracy and fidelity of the model fitting (Fig. 5 and ref. 12). The difference in the domain orientation is probably also due to the extra 18 residues insertion in the N-terminal region of FlgG compared with that of FlgE. Thus, a

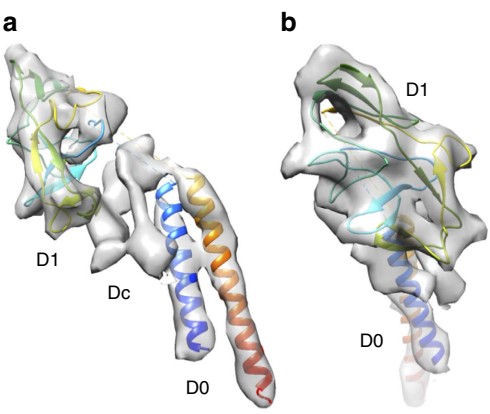

**Figure 5 | A magnified density map of a FlgG subunit to show the model fit.** (**a**) A side view showing domains D0, Dc and D1; (**b**) a view from outside of the rod by 90° rotation of the map and model shown in **a**. The N- and C-terminal α-helices, which form the inner core domain D0, are coloured blue and red, respectively, and the chain in domain D1 is coloured in rainbow. The domain connecting D0 and D1 is labelled Dc, as was designated in the hook structure[12].

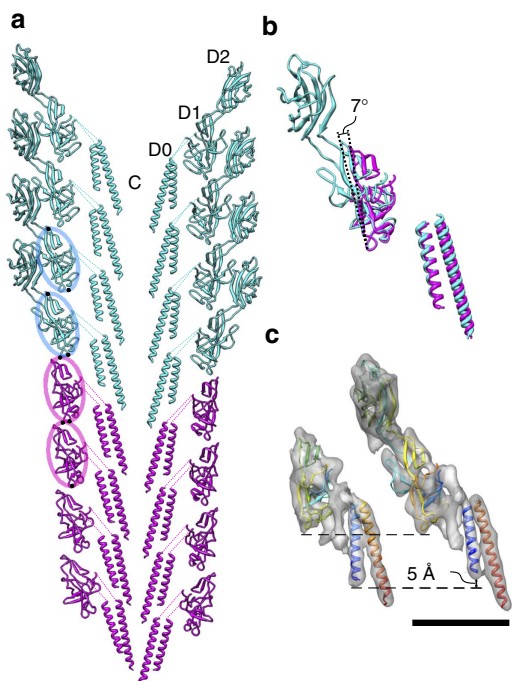

**Figure 6 | Comparison of the atomic models of the rod and hook in their direct axial connection.** (**a**) The rod (magenta) and hook (cyan) with their axial connection, shown in the longitudinal section though the axis. Domains D0, D1 and D2 are labelled with 'C' indicating the central channel. (**b**) Superposition of the FlgG (magenta) and FlgE (cyan) subunits, showing the ∼7° difference in the orientation of their D1 domains. (**c**) Comparison of the density maps showing that the N-terminal α-helix of FlgG is ∼5 Å longer than that of FlgE. Scale bar, 50 Å.

subtle difference in the quaternary structure produced by the 18 residues insertion must be responsible for the distinct mechanical properties of the two structurally very similar helical tubular assemblies of subunits with nearly identical folds.

**Straight polyhook formed by a FlgE mutant with insertion.** To examine our hypothesis that the rod is straight and more rigid as

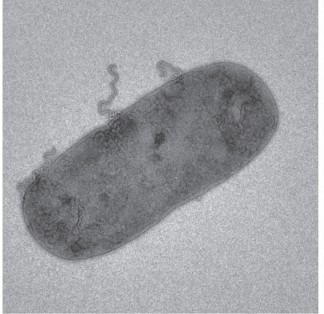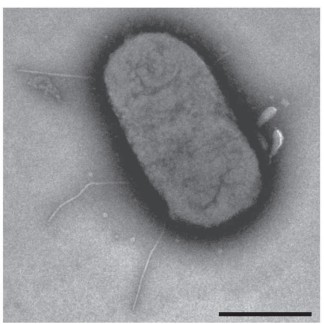

**Figure 7 | Electron micrographs of *Salmonella* cells with the wild-type and straight mutant polyhooks.** The polyhooks formed by wild-type FlgE are supercoiled and flexible (left) whereas those formed by a mutant FlgE with the insertion of 16 residues of the FlgG specific sequence between the FlgE residues Phe42 and Ala43 are straight and rigid (right). The cells were osmotically shocked and stained with 0.1% phosphotungstic acid. Scale bar, 1 μm.

a drive shaft than the hook as a universal joint by FlgG specific insertion of 18 residues not present in FlgE (Fig. 2a), we inserted this FlgG specific sequence into FlgE at the position between Phe 42 and Ala 43 to see if this actually makes the hook straight and rigid. We examined the effect of this insertion on a polyhook mutant produced by a mutation in FliK, a ruler protein that measures and determines hook length at around 55 nm (ref. 30). As shown in Fig. 7, while the polyhooks formed by wild-type FlgE are supercoiled and show a flexible nature, those formed by the FlgE mutant with 18 residues insertion are straight and rigid, demonstrating that the extra 18 residues in FlgG plays an important role in the formation of the rigid and straight rod to function as a drive shaft of the rotary motor.

## Discussion

Biological systems and their dynamic activities are all supported by complex interaction networks of macromolecules, such as protein, DNA and RNA, exchanging and converting small molecules, information and energy to exert specific functions for specific cellular activities. For each specific function, the amino acid sequence and 3D structure of a protein is designed and optimized by the evolutionary process over millions of years. Therefore, in general, proteins with distinct sequences and 3D structures are used for distinct functions. However, occasionally, biological systems take advantage of the versatility of protein design, where proteins of largely distinct functions are produced by small modifications in the amino acid sequence. In the present study, we have reported an interesting example of such kind. We show that two axial component proteins of the bacterial flagellum, FlgG and FlgE, which form the rod and the hook, respectively, share a high homology in their sequences and tertiary folds and assemble into very similar quaternary structures to form tubular structures, but produce distinct mechanical functions of the rod being rigid and the hook being flexible by a small insertion of amino acid residues.

We obtained structural evidence explaining how these two proteins of a nearly identical fold can produce different quaternary structures with distinct functions. Both FlgG and FlgE are considered to originate from a common ancestor, as they share 230 amino acids with 39% identity, and their tertiary structures are predicted to be nearly identical for the corresponding domains D0 and D1 (Fig. 2). The helical symmetries and repeat distances of the rod and hook are also nearly the same within the experimental error: an axial subunit rise of 4.13 Å and an azimuthal rotation of 64.8° for the rod, and an axial rise of

4.12 Å and a azimuthal rotation of 64.78° for the hook. However, the orientations of the D1 domains show a difference of ~7° between the rod and the hook, causing a distinct difference in their axial packing interactions. The D1 domains of FlgG subunits are oriented more upright, forming a tight axial packing interaction to build a rigid tubular structure of the rod to be a drive shaft. On the contrary, those of FlgE subunits are slightly more tilted, creating a small gap between the axially neighbouring subunits to allow compression and extension of each protofilament, thereby conferring a bending flexibility on the hook to work as a universal joint. The difference in their orientations by ~7° is probably caused by the insertion of 18 residues (residues 46–63) in FlgG, where many mutations have been identified to produce the polyrods[24]. Unfortunately, it is not possible to identify the main chain fold of this FlgG specific 18 residues insertion to build an atomic model due to the limited resolution of the density map, and therefore it remains unclear how this insertion modifies the orientation of domain D1. However, the above hypothesis has been proved to be correct by the formation of a rigid and straight hook by insertion of the FlgG-specific 18 residues into FlgE (Fig. 7).

The structure of the rod also revealed that domain D0, the innermost domain of the tubular structure, shows a very similar structure but distinct axial subunit packing in comparison with that of the hook. Domain D0 is composed of the terminal α-helices in both FlgG and FlgE. These terminal α-helices form a coiled coil in a similar manner to those of FliC in the flagellar filament, but they are significantly more tilted than those of FliC, with the longer C-terminal helix more tilted than the N-terminal one (Fig. 6a). In the hook structure, the axial subunit packing of these α-helices give rise to a gap of about 5 Å between subunits, with an overlap of about 10 Å between the two ends of the C-terminal α-helices (Fig. 4d), giving enough space for protofilament compression as well as stable intermolecular interactions. Thus, the structure and packing arrangement of FlgE subunits in the hook explains how the D0 domains allow the extension and compression of their axial array over 3–4 Å while maintaining the mechanical stability of the entire hook structure. On the contrary, the N-terminal α-helix of FlgG is about one pitch (~5 Å) longer than that of FlgE (Fig. 6c), thereby making a direct contact with the yet un-modelled connector domain Dc of the lower subunit (Fig. 4c) and preventing the FlgG subunit packing from changing their the axial repeat distance of the rod structure. So this difference in the structural design of the innermost domains of the rod and the hook is also responsible for their distinct mechanical properties, the former being rigid as a drive shaft and the latter being flexible in bending as a universal joint.

The structural and functional differences between the hook and the filament require a structural and mechanical adaptor to join them. FlgK and FlgL are the adaptor proteins that form the junction between these two tubular structures. Despite the functional differences between the rod and the hook, however, there are no junction proteins connecting them. Despite the ~7° difference in the orientation of the D1 domains in the rod and hook structures, the entire subunit structures and domain arrangements of FlgG and FlgE are probably similar enough to allow the intersubunit interactions between FlgG and FlgE to be maintained at the rod-hook junction (Fig. 6a). That is why the rod-hook connection is mechanically stable without any junction proteins.

The C-terminal α-helix faces the inner lumen while the N-terminal α-helix is located outside, facing domain Dc. This relative placement is the same as that in the atomic model of the filament[23]. The inner surface of the central channel of the filament is covered mainly by polar amino acids with one

positively charged residue, Arg 494. The polar nature of the surface appears to be advantageous for rapid diffusion of unfolded proteins through the channel because unfolded proteins have many hydrophobic side chains exposed. We built the atomic model of domain D0 in the polyrod structure by placing the hydrophobic residues of the predicted heptad repeat of the N- and C-terminal regions to face each other. As a result, the residues facing the central channel are mostly polar residues with three positively charged ones (Arg 238, Lys 245, Lys 256)

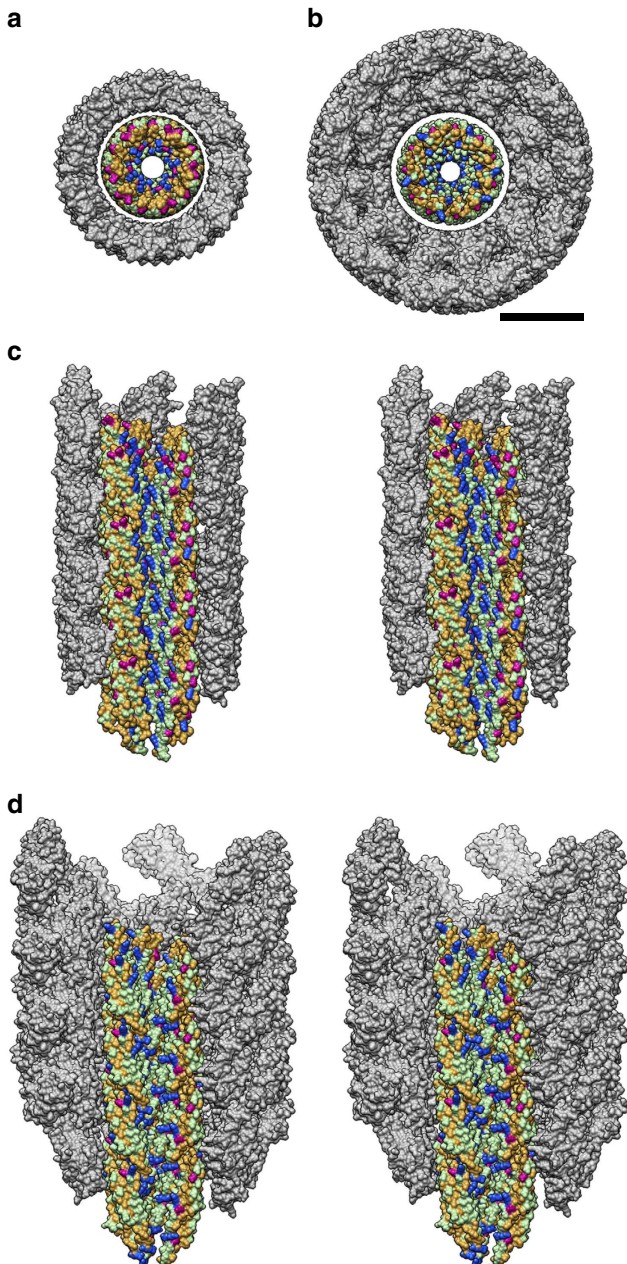

**Figure 8 | Distribution of residues with different characteristics in the central channel of the rod and hook.** Residues are colour coded as follows: positive charge (Arg, Lys, His), blue; negative charge (Asp, Glu), red; polar (Ser, Thr, Asn, Gln), light green; hydrophobic (Ala, Gly, Val, Leu, Pro, Ile, Cys, Met, Trp, Phe, Tyr), brown. Domains D1 and D2 are coloured grey. (**a,b**) End-on views of the rod (**a**) and hook (**b**). (**c,d**) Side views of the rod (**c**) and hook (**d**) in stereo. Three protofilaments on the near side of the tubular structures are removed to show the surfaces of the protofilament edge and the central channel. Scale bar, 50 Å.

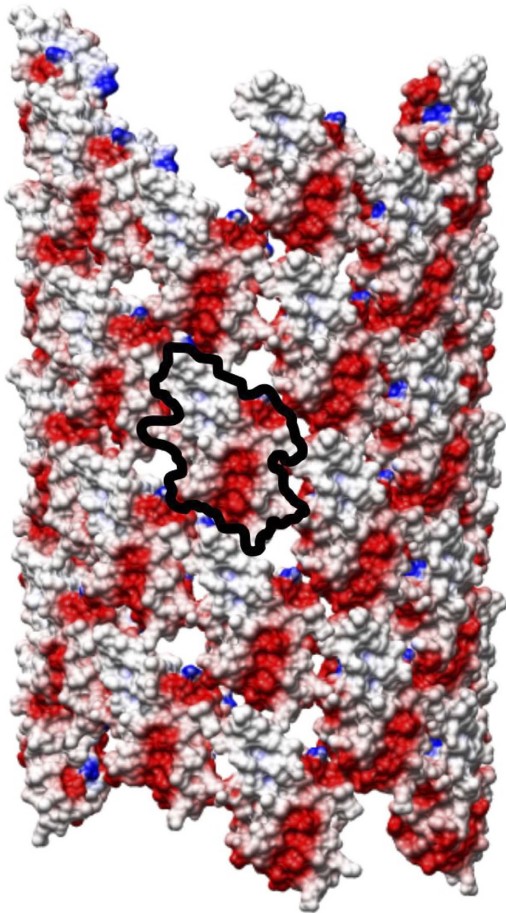

**Figure 9 | Electrostatic potential distribution of the polyrod surface.**
The negative potential is shown in red and the positive in blue. The potential distribution was calculated by Delphi[43] and shown by Chimera[29].

**Table 1 | Image collection and processing statistics.**

| | |
|---|---|
| Number of micrographs | 355 |
| Pixel size | 1.64 Å per pixel |
| Number of filaments | 734 |
| Total number of segmented images in initial selection | 11,206 |
| Total number of segmented images used in final map | 10,645 |
| The number of asymmetric units | 216,100 |
| Resolution (FSC = 0.5) | 7.4 Å |
| | |
| *Helical symmetry* | |
| Translation | 4.13 Å |
| Rotation | 64.75 |

*Yersinia* YscP, a FliK homolog, stays inside the central channel of the needle complex during needle assembly, partly in an α-helical conformation, and monitors the needle length as a ruler[31,32]. However, the size of the central channel of the rod and hook is only 13 Å, which is too small even for a completely unfolded chain of export substrate to go through if the FliK chain stays there as well. So, another, more favourable hypothesis is that FliK is intermittently exported for its N-terminal chain to act as a flexible tape measure to monitor the hook length during more frequent FlgE export for hook assembly[33].

The stoichiometries of the rod proteins FlgB, FlgC, FlgF and FlgG have been estimated to be about 6, 6, 6 and 26, respectively, as part of the study to determine the stoichiometries of the HBB component proteins by analyzing [35]S-labelled HBB complex, normalizing for the number of sulfur-containing residues of each protein, and converting the relative stoichiometry values into absolute ones based on the number of FlgE subunits in the hook (∼130) obtained from the hook length and its helical lattice[19]. Some of the estimated numbers are in good agreement with those determined structurally: 27.2 copies of FliF for the 26-fold rotational symmetry obtained by cryoEM image analysis of the FliF ring[34] and 12.6 copies of FlgK for 11 subunits identified by cryoEM image analysis of the hook-filament junction (F Makino, personal communication). The length of the thin portion of the proximal rod with a diameter of 6 nm is 4.5 nm, and if FlgB and FlgC assemble with the same helical parameters as those of the FlgG rod, the length 4.5 nm corresponds to about 11 subunits of FlgB and FlgC together. This is consistent with the biochemically-estimated stoichiometry 6 each for FlgB and FlgC. Interestingly, however, if the total number of FlgF and FlgG subunits that forms the thick distal portion of the rod is estimated from its length of about 20 nm, the subunit number is about 48, which is much larger than the biochemical estimation of 32 (6 + 26). Assuming that the stoichiometry of FlgF is correct as those of FlgB and FlgC, the number of FlgG subunits in the distal rod is ∼42, and this roughly corresponds to the model of the FlgG rod shown in Fig. 9, in which each of the 11 protofilaments mostly contains 4 FlgG subunits, and the length of the rod is about 20 nm.

Chevance *et al.*[24] found that FlgG rod assembly stops at the wild-type rod length for proper formation of the LP ring around it for the rod to penetrate the outer membrane for hook growth in the cell exterior. The L ring formation catalyses the removal of the rod cap, which is made of FlgJ and acts as the scaffold for rod assembly, to allow FlgD to form a new cap at the tip for hook assembly in cell exterior[35]. They assumed the FlgG rod to be composed of 26 subunits forming two stacks of the helical layer and discussed the stop-polymerization mechanism involving either conformational change or azimuthal rotation of FlgG subunits in the second layer to be large enough to prevent further polymerization. However, since the actual rod length is about 20 nm with 4 stacks of the helical layers, conformational or packing distortions of FlgG subunits if any must be more slowly

arranged linearly (Fig. 8). This model also supports the notion that the inner surface of the central channel is favourably designed for rapid diffusion of unfolded proteins for efficient translocation of flagellar proteins to the distal growing end for self-assembly. Interestingly, because the C-terminal α-helices of FlgG in the rod and FlgE in the hook are more tilted and protrudes toward the central channel than those of FliC in the filament, the diameter of the central channel in the rod and the hook are only ∼13 Å, much smaller than ∼20 Å of the flagellar filament[23] (Fig. 8a,b). The size of the central channel limits the size and conformation of export substrates, which must be unfolded to individual α-helices or an extended polypeptide chain to go through the channel smoothly. Even if proteins are completely unfolded to an extended polypeptide chain, more than two chains cannot be present within the 13 Å channel at the same time.

The hook length control is one of the intriguing questions in the regulatory mechanisms of flagellar assembly. It has been suggested that the extended N-terminal chain of FliK monitors the hook length during hook assembly, and the interaction of its C-terminal domain with the cytoplasmic domain of FlhB is responsible for the substrate specificity switch of the protein export apparatus to stop rod/hook protein export and start filament protein export upon the hook grows to 55 nm. How does FliK measure the hook length? One of the hypotheses proposed for needle length control in the assembly mechanism of the type III secretion system of pathogenic bacteria, which is homologous to the flagellar hook basal body, is that the N-terminal chain of

cumulative than what was previously discussed. A complete atomic model of the polyrod is necessary to identify the interactions around mutated residues of FlgG to understand the stop-polymerization mechanism. However, one possibility is that the rod polymerization is simply stopped or severely retarded by the outer membrane when the growing rod reaches its periplasmic surface because the polymerization ability of FlgG may not be strong enough to go beyond that point. FlgG mutants that produce polyrods[24] may have a stronger polymerization ability than wild-type FlgG to be able to overcome the physical limits on FlgG polymerization imposed by the outer membrane.

The outer surface of the polyrod has an electrostatic potential distribution strongly biased to negative (Fig. 9). Three negatively charged residues, D109, D154 and E203, are responsible for this surface potential. This negative-charge bias may be responsible for smooth and rapid rotation of the rod within the LP ring without any lubricant. If the inner surface of the LP ring is also negatively charged, the electrostatic repulsion can suppress the direct contact between the rod and the LP ring, reducing the friction between them. Since the electrostatic repulsion is inversely proportional to the square of the distance, and the gap distance between the rod and LP ring is likely to be around or less than 1 nm, the repulsive force must be quite strong. We can examine this hypothesis by replacing charged residues of FlgG as well as those of FlgH and FlgI and measuring the rotation speed of the mutant motors, but we need to wait until the LP ring structure becomes available at atomic resolution.

## Methods

**Sample preparation.** For preparation of the polyrod specimen, we used a polyrod strain of *Salmonella enterica* serovar Typhimurium, TH9709 [$\Delta fliK::tetRA$ $\Delta flgHI$ $flgG^*5667$(G65V)][24]. The polyrod was purified in the same way as the HBB isolation and purification as previously described[36]. The polyrod was suspended in a solution of 10 mM Tris–HCl (pH 8.0), 25 mM NaCl, 0.01 % Triton X-100 for cryoEM specimen preparation.

To insert the FlgG specific 18 residues sequence (YQTIRQPGAQSSEQTTLP), hereafter called 'RSS' (Rod Specific Sequence), between Phe 42 and Ala 43 of FlgE, we used the λ Red homologous recombination system[37] as described previously[38]. First, the *tetRA* genes were amplified by PCR using 5′-TGGCTTTAAGTCCGGT ACGGCATCATTTGCCGATATGTTCTTAAGACCCACTTTCACATT-3′ and 5′-TCCCCGCCACTTTTACGCCCAGCCCCACTTTGGAACCGGCCTAAGCA CTTGTCTCCTG-3′ as DNA primers. The reaction mixture was purified using a QIAquick PCR purification kit (QIAGEN). *Salmonella* SJW3076 strain harboring pKD46 (ref. 37) was grown in 5 ml of LB containing ampicillin and 0.2% L-arabinose at 30 °C until OD$_{600}$ had reached 0.6. The cells were washed three times with ice-cold H$_2$O and suspended in 50 μl of ice-cold H$_2$O. The 50 μl solution of cells were electroporated with 100 to 200 ng of purified PCR products using 0.1-cm cuvettes at 1.8 kV. Shocked cells were added to 1 ml LB and incubated for 1 h at 37 °C. Then, the cells were spread onto LA plates (12 g of Bacto agar (Difco) per liter of LB) containing tetracycline to generate the *flgE::tetRA* strain. Then, to remove the *tetRA* genes from the chromosome, the DNA encoding the RSS was amplified by PCR using 5′- GGCTTTAAGTCCGGTACGGCATCATTTGCCGA TATGTTCTATCAGACCATCCGCCAGCC-3′ and 5′- TCCCCGCCACTTTTA CGCCCAGCCCCACTTTGGACCGGCAGGCAGCGTCGTCTGCTCGG-3′ as DNA primers. The *flgE::tetRA* strain carrying pKD46 was electroporated with purified PCR products. After 1-hour incubation at 37 °C, cells were spread onto tetracycline-sensitive plates and incubated at 42 °C overnight to generate a *Salmonella flgE::RSS* strain (MME1001). The *flgE::RSS* allele was confirmed by DNA sequencing. DNA sequencing reactions were carried out using BigDye v3.1 as described in the manufacturer's instructions (Applied Biosystems), and then the reaction mixtures were analysed by a 3,130 Genetic Analyzer (Applied Biosystems). The Δ*fliK::tetRA* allele derived from the *Salmonella* Δ*fliK* Δ*fliL* Δ*fliM* strain (NMEK001)[39] were transduced into the *Salmonella* MME1001 and SJW1103 (wild type)[40] strains by P22-mediated transduction to create the *flgE+RSS* Δ*fliK::tetRA* (MMEK001) and Δ*fliK::tetRA* (MMK001) strains, respectively, which produce polyhooks composed of FlgE with and without RSS insertion, respectively.

**Electron cryomicroscopy.** A 4 μl sample solution was applied onto a Quantifoil grid (R1.2/1.3, Quantifoil Micro Tools GmbH), and the grid was plunge-frozen into liquid ethane using a fully automated vitrification device (Vitrobot, FEI). The specimen was observed at a specimen temperature of around 50 K using a JEOL JEM-3200FSC electron microscope, equipped with a liquid-helium cooled specimen stage, an Ω-type energy filter and a field-emission electron gun operated

at 200 kV. Zero-loss images (10 eV) were recorded on a 4 k × 4 k 15 μm-per-pixel slow scan CCD camera (TemCam-F415MP, TVIPS). The electron dose was ∼ 20 electrons per Å$^2$. The magnification of × 91,463 and the pixel size of 1.64 Å per pixel were determined by measuring the 23.0 Å layer line spacing from images of tobacco mosaic virus (TMV) recorded in the same CCD frames with polyrods and polyhooks.

**Image analysis.** Image analysis was performed as previously described[12]. Polyrod images were segmented into 11,206 segment images of 512 × 512 pixels with a step size of 51 pixels along the helical axis using EMAN's boxer program[41]. The images were then high-pass filtered (285 Å) to remove a low-spatial frequency undulation of density, normalized and cropped to 320 × 320 pixels. An averaged power spectrum was generated by computing the Fourier transform of each segment box and then adding up the intensities of these spectra together. The power spectrum thus generated is invariant with respect to any image shifts and therefore allows reliable determination of the helical symmetry and repeat distance of the polyrod structure.

Further image processing was carried out with the SPIDER package[42] and IHRSR method[28]. To reduce the search range in refinement and speed up the alignment process, segment images were processed with the coarse alignment procedure. The axis of the polyrod was aligned to the midline of the image using one-dimensional projection matching between segment images and reference images. A negative temperature factor of 200 Å$^2$ was applied to the final 3D image reconstruction, and the 3D image was filtered at 7.4 Å resolution. Image collection and processing statistics are summarized in Table 1. The homology modelling of FlgG$_{91–222}$ and model fitting into the cryoEM density were carried out as described in Results.

**Electron microscopy of polyhooks on osmotically shocked cell.** A 50 μl overnight culture solution was inoculated to 5 ml of LB and cultured at 37 °C until OD$_{600}$ had reached 0.6. The culture solution was centrifuged at 7,000 r.p.m. for 10 min at 4 °C, and the cells in the pellet were suspended in a solution containing 0.75 M sucrose, 150 mM Tris base and 10 mM EDTA and incubated at 37 °C for 20 min. The cell solution was centrifuged at 14,000 r.p.m. for 10 min at 4 °C, and the cells in the pellet were suspended in 20 ml of ice cold water for osmotic shock. Then, 1 μl (5 units) of DNAase-I (Takara) and 1 ml of 0.1 M MgSO$_4$ were added, and after incubation at 30 °C for 20 min, it was centrifuged again at 14,000 r.p.m. for 20 min at 4 °C. The cells in the pellet was suspended in 200 μl of water and used for EM observation of polyhooks. A 4 μl solution of the osmotically shocked cells was applied on an EM grid, and the cells were negatively stained with 0.1% phosphotungstic acid solution (pH 6.0) by repeating blotting and staining three times. A JEOL JEM-1011 electron microscope was operated at 100 kV to observe the cells on the EM grid at a magnification of × 7,500.

**Data availability.** The reconstructed density was deposited to Electron Microscopy Data Bank with accession code EMD-6683 and the atomic coordinate to Protein Data Bank with PDB ID code 5WRH. The data that support the findings of this study are available from the corresponding author upon request.

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

## Acknowledgements

We thank Edward Egelman for his kind help in the use of the IHRSR method in early stage of this study, and Izumi Tokita, Masaaki Urabe and the JEOL company for maintaining the electron cryomicroscope at the best conditions. T.F. was a research fellow of the Japan Society for the Promotion of Science. T.K. was funded by the Global COE Program to the Graduate School of Frontier Biosciences, Osaka University. This work was supported by JSPS KAKENHI Grant numbers JP25711010 to T.F., JP24570131 to T.Miyata. and JP21227006 and JP25000013 to K.N., MEXT KAKENHI Grant Number JP24117004 to T. Minamino, and NIH grant GM056141 to K.H.

## Author contributions

T.F. made various improvements in the cryoEM method, performed all the experiments and developed all image analysis programs used here; T.K. set up and managed the cryoEM and computing facilities for image analysis and fine-tuned the parameters to make best possible use of them; K.D.H., T. Miyata and T. Minamino made a polyhook mutant strain of *Salmonella* with a FlgE mutant with an insertion of the FlgG specific sequence and observed the polyhook by electron microscopy of osmotically shocked cell; F.F.V.C. and K.T.H. designed and constructed strains to produce polyrods; K.N. supervised the whole project; and T.F. and K.N. wrote the paper based on discussion with the other authors.

## Additional information

**Competing financial interests:** The authors declare no competing financial interests.

**Publisher's note**: 

