## [Peer Review File · Nature Communications]

Reviewers' Comments:

Reviewer #1 (Remarks to the Author)

Fujii et al. determined the structure of the rod component of Salmonella flagellum and provided a hypothesis on why there is a difference in rigidity of the rod and the hook. Even though flagellum is a paradigm for studying molecular self-assembly and it is interesting to learn how similar molecular structures could result in distinct mechanical properties, the current quality of this work does not provide sufficient interest to the broad authorship of Nature Communication for at least two reasons: (1) the lack of resolution of the EM map prevents building a complete atomic model of the rod; (2) the cause of difference in rigidity between the rod and the hook is very speculative and further work, for example, modifying the length of N-terminal, is needed to validate it. I therefore recommend it to be published in a more specialized journal.

Some minor concerns:

- (1) IHRSR resulted in the converged helical symmetry basically the same as the initial value. Did the authors try different initial values to boost the confidence of the result?
- (2) "A common fold used for two distinct functions" in Discussion is basically a redundancy of "Structural comparison of the rod and hook" in the Results.
- (3) In general, Discussion is very long and many parts seem irrelevant to the work, e.g, page 11, lines 320 - 334, discussion on the hook length control.
- (4) Page 3, line 41. C-ring is mentioned to be 45 nm wide, but the work by Chen et al, 2011 "Structural diversity of bacterial flagellar motors" showed a variety of diameters. Maybe the name of the specific organism is needed?

Reviewer #2 (Remarks to the Author)

In this work, Fujii and coworkers address an important problem in structural biology: How two seemingly identical protein folds achieve different biological functions? Focusing on the rod and hook proteins FlgG and FlgE, and using a 7 Å resolution density map from cryoEM, the authors put forth a compelling argument of how subtle structural changes in the helical molecular assembly result in changes in flexibility and eventually biological function. This work is of great

interest to the broad structural biology community, but also to more specialized areas such as cryoEM and bacterial locomotion. From a technical point of view, the paper is solid, and the conclusions are largely supported by the results presented in the paper. I therefore suggest publication of the manuscript as is, as I think it will be of great value to the readership of Nature Communications.

Response to the Referees' comments

To Referee #1:

Fujii et al. determined the structure of the rod component of Salmonella flagellum and provided a hypothesis on why there is a difference in rigidity of the rod and the hook. Even though flagellum is a paradigm for studying molecular self-assembly and it is interesting to learn how similar molecular structures could result in distinct mechanical properties, the current quality of this work does not provide sufficient interest to the broad authorship of Nature Communication for at least two reasons: (1) the lack of resolution of the EM map prevents building a complete atomic model of the rod; (2) the cause of difference in rigidity between the rod and the hook is very speculative and further work, for example, modifying the length of N-terminal, is needed to validate it. I therefore recommend it to be published in a more specialized journal.

Regarding point (1), although the resolution determined by the FSC = 0.5 is 7.2 Å, the homology model of domain D1 of FlgG was fitted into the density map as a rigid body, and all the secondary structures including β -sheets, β -hairpins and loops were clearly identified in the density map. The lengths of the rod-shaped densities to which the terminal α -helices were fitted also agreed well with those predicted from their amino acid sequences. Thus, we can assure that the resolution and quality of the density map is sufficiently high for accurate and reliable model building under such strong stereochemical constraints. Therefore the model is accurate enough especially for the discussion of the orientation of domain D1 and the lengths of terminal α -helices.

Regarding point (2), we constructed a *flgE* mutant by inserting the 18 residues that are present only in the FlgG sequence (residues 46 – 63) to see if the hook becomes straight and rigid as the FlgG rod. The mutant cells indeed produced straight hooks as shown in the figures below. The polyhooks produced by the *flgE* mutant cells with deletion of the hook ruler protein FliK were also straight and rigid. Thus, we have solid evidence that validates the cause of the difference in rigidity between the rod and hook. However, we cannot include these results in this manuscript because this study was carried out by one of my graduate students for his PhD thesis study, and the manuscript is in preparation to be published elsewhere.

The flagellar hook basal bodies isolated from the wild-type cell (left) and *flgE* mutant (right).

Polyhooks extending from the wild-type cell (left) and the *flgE* mutant (right).

Fluorescence microscopic images of the wild-type (left) and *flgE* mutant (right). The wild-type cells swim smoothly by forming a filament bundle behind the cell body, but the *flgE* mutant cannot swim because the individual filaments are extending in different directions due to the rigid and straight hooks.

Some minor concerns:

(1) IHRSR resulted in the converged helical symmetry basically the same as the initial value. Did the authors try different initial values to boost the confidence of the result?

Yes, we tried a range of initial values, $4.12 \pm 0.2 \text{ \AA}$ for axial rise and $64.78^\circ \pm 0.5^\circ$ for azimuthal rotation, and they all converged to the same values. Because four layer lines that were clearly visible in the Fourier transform of the rod were at the same axial positions as those of the hook, there was no ambiguity in the helical parameters. We inserted a phrase in the relevant sentence to make sure it is clear to readers.

(2) "A common fold used for two distinct functions" in Discussion is basically a redundancy of "Structural comparison of the rod and hook" in the Results.

Some part of Discussion may be somewhat redundant but we believe it is necessary for the discussion to be comprehensible for the broad readership.

(3) In general, Discussion is very long and many parts seem irrelevant to the work, e.g, page 11, lines 320 - 334, discussion on the hook length control.

We believe that describing the functional implications of the rod structure is important part of this work. This part of Discussion describes one of the important implications of the rod channel size as the path for the export of proteins, such as FlgG, FlgE and FliK, for the rod and hook assembly, and therefore it is quite relevant to the work. The mechanism of hook length control by the ruler protein FliK is still under active debate as to whether one FliK molecule stays inside the central channel to measure the hook length while FlgE molecules are exported through the central channel for hook assembly or a small number of FliK molecules are exported alternatively with many FlgE molecules to measure the hook length occasionally. The inner diameters of the channels of the rod and hook give very important clues that dismiss the former model.

(4) Page 3, line 41. C-ring is mentioned to be 45 nm wide, but the work by Chen et al, 2011 "Structural diversity of bacterial flagellar motors" showed a variety of diameters. Maybe the name of the specific organism is needed?

We inserted a phrase "In Gram-negative bacteria, such as *Salmonella enterica* and *Escherichia coli*," in the beginning of the sentence to specify the organism.

To Referee #2:

In this work, Fujii and coworkers address an important problem in structural biology: How two seemingly identical protein folds achieve different biological functions? Focusing on the rod and hook proteins FlgG and FlgE, and using a 7 \AA resolution density map from cryoEM, the authors put forth a compelling argument of how subtle structural changes in the helical molecular assembly result in changes in flexibility and eventually biological function. This work is of great interest to the broad structural biology community, but also to more specialized areas such as cryoEM and bacterial locomotion. From a technical point of view, the paper is solid, and the conclusions are largely supported by the results presented in the paper. I therefore suggest publication of the manuscript as is, as I think it will be of great value to the readership of Nature Communications.

Thank you very much for your appropriate evaluation of this study.

Reviewers' comments:

Reviewer #1 (Remarks to the Author):

I am still convinced that the current state of this paper does not have high impact since the main mechanistic model for the difference in rigidity between the rod and the hook is still presented as a speculation, and therefore the paper is only suitable for a more specialized journal.

Since the authors have shown that a flgE mutant with insertion of an extra residues of the FlgG sequence produces straight hooks, I suggest that the authors include this solid validation to their paper to make it impactful and suitable for the broad audience of Nature Communications.

Response to the Referees' comments

To Referee #1:

I am still convinced that the current state of this paper does not have high impact since the main mechanistic model for the difference in rigidity between the rod and the hook is still presented as a speculation, and therefore the paper is only suitable for a more specialized journal. Since the authors have shown that a flgE mutant with insertion of an extra residues of the FlgG sequence produces straight hooks, I suggest that the authors include this solid validation to their paper to make it impactful and suitable for the broad audience of Nature Communications.

Thank you for your kind suggestion. We revised our manuscript by including experimental data as Fig. 7 to demonstrate that insertion of the FlgG specific sequence into FlgE actually produces straight hooks. As shown in EM images of polyhooks in Fig. 7, while the polyhooks formed by wild-type FlgE are supercoiled and show a flexible nature, those formed by the FlgE mutant with 18 residues insertion are straight and rigid, demonstrating that the extra 18 residues in FlgG are responsible for the formation of the rigid and straight rod to function as a drive shaft of the rotary motor. This result is described in the last section of Results in page 9.